# Raw Nav-merge Seismic Data to Subsurface Properties with MLP-based Multi-Modal Information Unscrambler

**Aditya Desai**[*]
Rice University
apd10@rice.edu

**Zhaozhuo Xu**[*]
Rice University
zx22@rice.edu

**Menal Gupta**
Shell Exploration & Production Company
Menal.Gupta@shell.com

**Anu Chandran**
Shell India Markets Private Limited
Anu.Chandran@shell.com

**Antoine Vial-Aussavy**
Shell Exploration & Production
a.vial-Aussavy@shell.com

**Anshumali Shrivastava**
Rice University and ThirdAI Corp.
anshumali@rice.edu

## Abstract

Traditional seismic inversion (SI) maps the hundreds of terabytes of raw-field data to subsurface properties in gigabytes. This inversion process is expensive, requiring over a year of human and computational effort. Recently, data-driven approaches equipped with Deep learning (DL) are envisioned to improve SI efficiency. However, these improvements are restricted to data with highly reduced scale and complexity. To extend these approaches to real-scale seismic data, researchers need to process raw nav-merge seismic data into an image and perform convolution. We argue that this convolution-based way of SI is not only computationally expensive but also conceptually problematic. Seismic data is not naturally an image and need not be processed as images. In this work, we go beyond convolution and propose a novel SI method. We solve the scalability of SI by proposing a new auxiliary learning paradigm for SI (Aux-SI). This paradigm breaks the SI into local inversion tasks, which predicts each small chunk of subsurface properties using surrounding seismic data. Aux-SI combines these local predictions to obtain the entire subsurface model. However, even this local inversion is still challenging due to: (1) high-dimensional, spatially irregular multi-modal seismic data, (2) there is no concrete spatial mapping (or alignment) between subsurface properties and raw data. To handle these challenges, we propose an all-MLP architecture, Multi-Modal Information Unscrambler (MMI-Unscrambler), that unscrambles seismic information by ingesting all available multi-modal data. The experiment shows that MMI-Unscrambler outperforms both SOTA U-Net and Transformer models on simulation data. We also scale MMI-Unscrambler to raw-field nav-merge data on Gulf-of-Mexico to obtain a geologically sound velocity model with an SSIM score of 0.8. To the best of our knowledge, this is the first successful demonstration of the DL approach on SI for real, large-scale, and complicated raw field data.

---

[*]indicates equal contribution

35th Conference on Neural Information Processing Systems (NeurIPS 2021).

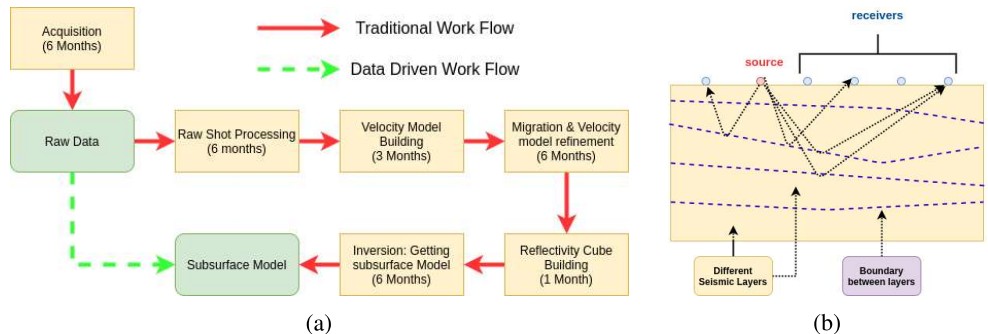

(a)                           (b)

Figure 1: **Left**:Traditional workflow involved in seismic processing of raw data to get a subsurface model. It requires approximately over an year time to generate a subsurface image. Data-driven workflows try to bypass this tedious process by directly predicting subsurface model from raw data. **Right**: Reflection Seismology for collecting seismic data: Mechanical Perturbation (sound wave) created at source, propagates through layers of earth and reflects subsurface boundaries. The receivers record amplitude time-series information.

# 1 Introduction

Seismic reflection data records the amplitude of reflected waves from the subsurface of the earth. Geophysicists perform Seismic Inversion (SI) on this data to generate the earth's subsurface properties. However, seismic data is known to be notoriously difficult to analyse. The sheer scale of the seismic data is extremely large. For example, a survey over a region (100Km $\times$ 100Km) inside gulf of mexico generates hundreds of terabytes of data. Moreover, the data is highly complex. There is no concrete spatial mapping between subsurface properties and observed data. Furthermore, each recording captures superimposed information of the entire subsurface. These characteristics make information unscrambling a hard task in SI. Traditional approaches are physics-driven, which require an extensive amount of human expertise and time. Recently, there is an increasing interest in using deep learning (DL) to solve SI efficiently. However, current DL research in SI restricts to synthetic data and is difficult to scale to real SI settings. In this work, we solve the scalability of SI by proposing a new auxiliary learning paradigm for SI (Aux-SI). In this paradigm, we break the SI into local inversion tasks, which predicts each small chunk of subsurface properties using surrounding seismic data. Aux-SI combines these local predictions to obtain the entire subsurface model. However, even this local inversion is challenging. Firstly, the geometry of data acquisition is highly irregular. This, along with the fact that data is high dimensional (dimension $\geq$ 5) puts natural choices like CNN at a disadvantage. Secondly, the seismic data is multi-modal as it contains different types of information like amplitude recordings (time-series), locations of sources and receivers (geo-locations), instrument sensitivity measurements, the actual signature of disturbance, time of recordings, etc. To handle these challenges, we propose a neural network via only Multi-Layer Perceptron(MLP), dented as "Multi-Modal Information Unscrambler " (MMI-Unscrambler). MMI-Unscrambler can efficiently ingest varying types of data and is robust to spatial irregularities. Moreover, it is an all-MLP network. We show in our experiments MMI-Unscrambler can invert real and large scale raw-field nav-merge data to produce geologically sound velocity images with high SSIM scores.

**Expensive seismic processing workflows:** Traditional state-of-the-art methods of processing seismic data consume significant resources and time [1]. Figure 1a shows broader steps in traditional workflow commonly used in the "Big Oil" industry. It is well known [2] that each of the sequential steps in traditional workflows takes several months. This time inefficiency is primarily due to their dependence on human expertise to process and examine vast amounts of raw data. With the increasingly comprehensive and complex seismic data generated by advanced sensing technology, traditional workflows prove to be more expensive than ever. Hence, there is an urgent need to devise techniques to complement the existing process with automated technology to speed up the overall process.

**Promise of deep Learning (DL):** The data-driven approaches [3, 4, 5, 6] are progressively sought alternative to improve seismic workflows' efficiency. Data-driven deep learning (DL) approaches aim to learn a non-linear mapping from the raw seismic data to the final subsurface image evading most of the time-consuming sequential steps in traditional workflows. With the fast inference of DL models, data-driven approaches can potentially reduce workflows' duration from months to minutes. Therefore, it is not surprising that research for DL-based workflows in seismic processing are getting the attention and investment they deserve [7, 8, 9, 10, 11, 12, 13, 14, 15, 16].

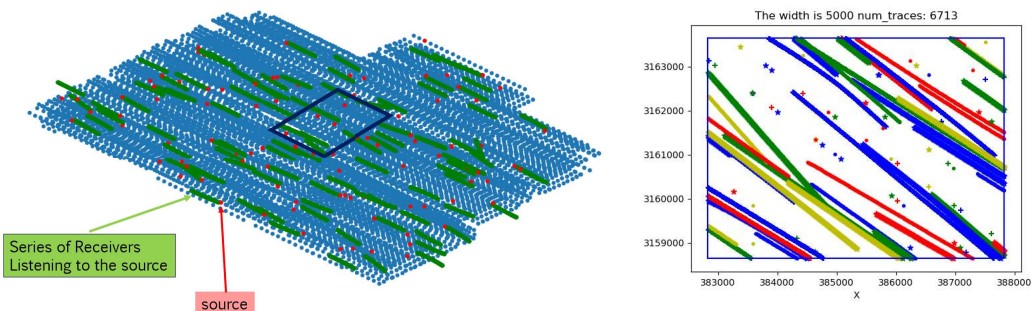

Figure 2: Irregular Acquisition in real SDI. **Left**: A top view of a real survey. The survey area is large (around 100Km × 100 Km) and the streamer (in marine survey) moves across the area to collect data. The blue dots represent the source points in the survey. Red dots and nearby green segments refer to a pair of source and series of receivers. We focus on a particular area to predict the subsurface properties. **Right**: Details at the acquisition over a 5Km x 5Km area. Different colors and symbols are used to identify pairs of source and receiver lines. The real acquisition is highly irregular emphasizing the oversimplification of data used in current convolution models.

**Convolutions in data-driven processing and limited success of DL in real seismic processing:** Treating raw seismic data as images is standard in the oil industry and geophysics community. Undoubtedly, image is the most convenient and interpretable representation of complex data. The recent data-driven research inherits the image bias from traditional practices. Inspired by the success of convolutions in computer vision, it attempts to process the raw seismic data as images and apply convolution-based networks to ingest them. [13, 7, 17, 10, 11, 12, 14]. Two characteristics of data put CNN-based SI at a great disadvantage 1) Real seismic data is at least 5 dimensional. Applying 5D convolutions to the extensive seismic data is computationally infeasible. 2) Real seismic data collection is highly irregular and unstructured (see Figure 2) and CNNs require a structured form of data. We can potentially interpolate traces to bring them on a regular grid. However, this would be a sub-optimal way to use seismic data. It is important to note that seismic data is not an image naturally, and hence it is an interesting question whether we should even try to process it as an image. Our results show that we can use only MLP blocks with MMI-Unscrambler architecture and get better results than CNN-based U-net on simulation data. Also, we obtain good results on real seismic data where applying U-net is computationally prohibitive. To the best of our knowledge, most CNN-based SI approaches are prohibitive to real seismic settings. Current research in CNN-based SI focuses on simulated data on 2D subsurface data with regular acquisition geometry (see Figure 6b). We believe the gap between simulation and practice is too wide to extend these methodologies to practical environmental applications.

**Major contributions in the paper:**

- We propose Aux-SI approach. This redefines the seismic inversion problem on terabytes of data into the auxiliary task of predicting small blocks of property models from their contextual traces. Thus, we make the training and inference of deep learning models on real data feasible.
- We introduce MMI-Unscrambler architecture which (1) incorporate multiple modes of information seamlessly (2) scale to higher dimensions of data with the addition of more metadata. (3) is robust to highly irregular acquisition geometry, and (4) is resistant to over-fitting.
- We believe, this is the first-ever data-driven approach using deep learning that can scale to real data and provide geologically sound results. In the experiments on real proprietary data from the survey on the Gulf of Mexico, we achieve a SSIM similarity (0-worst, 1-best) of over 0.8 on predicting 3D velocity model cubes.
- It is interesting to note that we are able to improve over the state-of-the-art U-Net for SI using MMI-Unscrambler architecture with only MLP blocks.

## 2  Background

### 2.1  Seismic Data and Associated Applications

**Reflection Seismology[18]:** Reflection Seismology is used to collect information about the subsurface so that geophysicists can construct a property model. As shown in Figure 1b, a source and a set

of receivers are positioned on the surface of the earth. A mechanical perturbation is created using air guns at the source. The pressure waves thus generated travel through the subsurface. Waves refract and reflect at different layer boundaries in the subsurface. The reflected pressure waves that reach the receivers are recorded to generate time-series data. Hence, each source shot and receiver pair is associated with a one dimensional time series data called the *trace-data* ($\mathcal{D}(t)$) and locations of source and receivers called as *acquisition geometry* ($\mathcal{A}(t)$). These terms are formally defined below. Apart from these, there is additional meta-data associated with each trace, such as the signature of disturbance created, instrument sensitivity of receivers, record times (the time when this recording was made), etc.

**Definition 1 (Trace data ($\mathcal{D}(\mathbf{t})$)).** *Let the receiver located at a location $q \in R^3$ listen for time duration $s$ seconds and record at a frequency $f$ (times per second), a trace received by the receiver is a sequence $\{u(s_i, x)\}_{i=1}^{fs}$ where $u(s_i, q) \in R$ is the acoustic pressure sensed at $q$ at the ith recording.*

**Definition 2 (Acquisition geometry of a trace ($\mathcal{A}(t)$)).** *The location of source and receiver associated with a particular trace-data, together constitute the acquisition geometry of the trace. Formally $\mathcal{A}(t) = (q_{src}, q_{rev})$. where $q \in R^3$ is a 3-dimensional location in space.*

In a typical survey, a source is located on the streamer, and long cables with equidistant receivers are attached to the streamer. The streamer moves over the area of interest and collects a large set of traces for different (source, receiver) pairs. The set of all (source, receiver) locations associated with the collected traces are collectively referred to as the acquisition geometry ($\mathbb{A}$). Also, the set of all traces collectively is referred to as raw data, $\mathbb{R}$

**Seismic Inversion(SI):** The final aim of seismic processing is to obtain a subsurface property model. This problem is referred to as the seismic inversion problem stated formally below. This property model can then identify the rock and fluid types in the subsurface.

**Problem 1 (SI).** *Given a region of interest $S \subset R^d$ and $\mathbb{R}$ (raw data), Seismic Inversion(SI) is defined as the problem of getting the property model $\mathcal{P}$ where $\mathcal{P} : S \rightarrow R$ is a mapping from each point in the subsurface to the value of the property of interest.*

The property here can be any set of elastic properties, intermediate velocity or reflectivity index, etc. We use the term *property* in the most general manner. The subsurface region of interest, $S$, in Problem 1 can be 2D or 3D space. Accordingly, the inversion problems are categorized as 2D or 3D problems. 2D problems generally appear in synthetically generated datasets.

## 2.2 Relevant Prior Work

**Traditional approaches** [19]: Traditionally, SDI for seismic workflows comprises a series of sequential steps involving computer-assisted human analysis. Generally, these steps are physics-driven. An example of workflow with approximate timelines is shown in Figure 1a. Raw shot processing cleans up the data by removing unintended noises that appear in the data. It includes sequential steps like de-noising, de-signature, de-multiple etc [19, 20, 21, 22]. The clean data is used to generate a velocity model of the subsurface using techniques of tomography [23, 24] or full-waveform inversion [25, 26]. The velocity model is then refined iteratively by migrating the waves, and computing errors like residual move out [27]. Once a refined velocity model is obtained, it is used to build a reflectivity cube of the subsurface. This cube is then inverted to obtain elastic properties of the subsurface (subsurface property model). Types of rocks and fluids in the subsurface are identified by analyzing the property model. Each of these steps require the involvement of domain experts to guide the inversion to an effective and applicable solution. These procedures extend the overall process to as lengthy as a year. For a detailed discussion on these steps, we refer the readers to [19].

**Recent DL approaches:** Though the application of data-driven methods to seismic processing dates back to the 1990s [28], there has been a renewed interest since the success of deep learning [29]. The data-driven methods [13, 11] aim to utilize the strength of deep learning [29] to exploit the available extensive complex seismic data fully. Most existing methods formulate Problem 1 as an image segmentation problem and utilize convolutional neural networks (CNN) [30, 31]. In this formulation, the traces are combined as multi-channel images, and the corresponding subsurface property model is regarded as a segmentation mask. Then CNN architectures such as U-Nets[10, 11, 12] or encoder-decoders [13, 14] are applied to learn a non-linear mapping between image-mask pairs. As stated earlier, these approaches do not scale to real data.

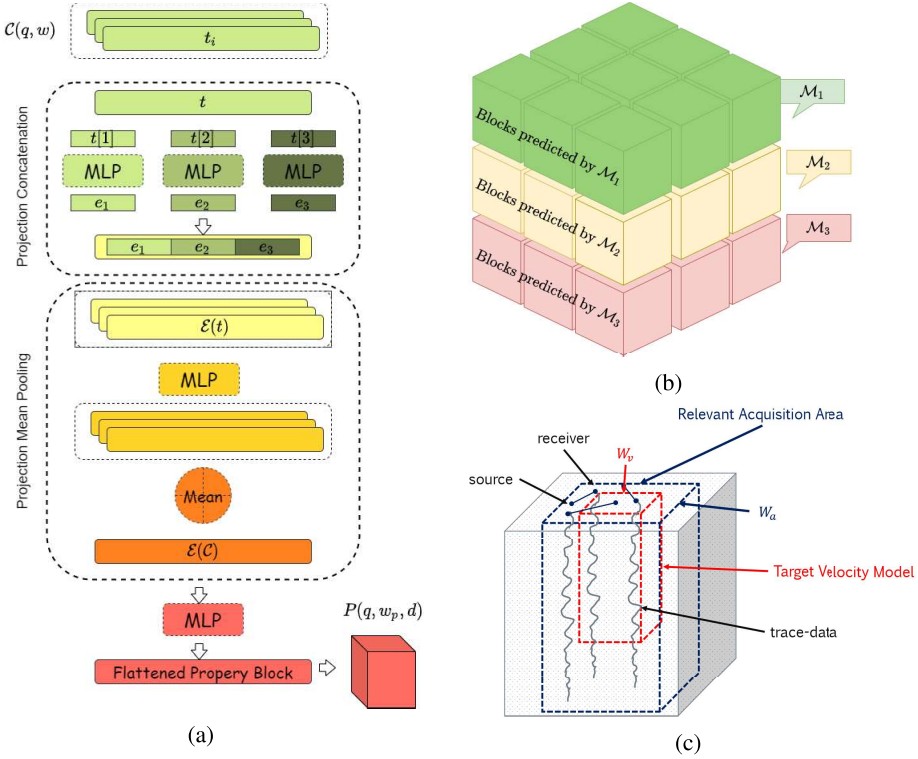

Figure 3: (a) General MMI-UnscramblerArchitecture. See the MMI architecture applied to velocity inversion at figure 7. (b) Creating entire property model using different models at different depths with each model traversing entire horizontal surface predicting one block at a time (c) Construction of context for a particular property model block placed near surface of earth

## 3 Our Approach to Seismic Inversion with Deep learning

### 3.1 Aux-SI: Viewing Seismic Inversion as Machine Translation

Applying deep learning to SI entails finding a map from several terabytes of raw-field data to several gigabytes of property-model of the subsurface. We call this problem global-inversion. It is prohibitive to train a single mapping from entire raw data to property model. We solve the global-inversion problem by proposing an auxiliary learning paradigm, namely Aux-SI. In Aux-SI, we break down the problem of global-inversion into multiple local-inversion tasks. The local-inversion tasks are much smaller in scale and allow efficient training of DL models. The solutions of these local inversion tasks can then be combined (stitching) to obtain a solution for the global-inversion problem. One can draw an analogy to state-of-the-art NLP solutions for machine document translation. Instead of learning a complete mapping between two documents (global-translation), DL models are learned to solve the local-translation problem: predict the next word considering a small context around it. The Local-translation model is then used to predict the entire document one word at a time. Aux-SI is inspired by this approach.

We first discuss the formulation of the local-inversion task. The target of this task is a small 3D chunk in the complete property model. Let $\mathcal{P}(q, w_p, d) \in R^{w_p \times w_p \times d}$ denote a chunk of property model of size $w_p \times w_p$ in area and $d$ in depth at a location $q \in R^3$ in subsurface. The input to this task is all the seismic traces that are found in a local area around $\mathcal{P}(q, w_p, d)$. We call this as the context of the block. This is illustrated in Figure 3c. Formally, context is defined as

**Definition 3 (Context $\mathcal{C}(q, w)$).** *The context, $\mathcal{C}(q, w)$, at a point $q$, parameterised by width $w$ is defined as the set of traces whose acquisition (source and receiver locations) lies inside $Area(q, w)$, i.e. the area $w \times w$ centered at $(q.x, q.y)$*

$$\mathcal{C}(q, w) = \{t | t \in \mathbb{R}, \mathcal{A}(t) \in Area(q, w)\}$$

*where $\mathbb{R}$ is the entire raw data.*

DL based solutions to the local inversion problem would learn a mapping $\mathcal{M}$, between $context(q, w_a)$ and property model block $\mathcal{P}(q, w_p, d)$. The scale of this learning problem is much lesser than the

original problem, making it feasible to train a model efficiently. In section 3.2, we will discuss the MMI-Unscrambler architecture to solve local-inversion.

Theoretically, the information about a particular $\mathcal{P}(q, w_p, d)$ is captured in all the traces across the entire survey. However, the impact of $\mathcal{P}(q, w_p, d)$ is strongest on traces near it, and the effect wanes out as we go to farther traces. Choosing the size of the context is a standard accuracy-efficiency trade-off. Nevertheless, geophysics dictates that the effect of property model chunks at deeper locations would be captured in traces located further away from the block. Hence, in Aux-SI, we propose to learn different mappings for different depths and choose wider contexts for deeper blocks. To obtain the complete property model, we divide the complete subsurface into chunks as shown in Figure 3b. Each chunk, $\mathcal{P}(q, w_p, d)$ is then predicted using a model $\mathcal{M}(q.z)$ and a context $\mathcal{C}(q, w(q.z))$. "q.z", which denotes depth of point q, in previous notation are used to make dependence explicit. This is a simple way to combine outputs of local inversion to get combined velocity. More evolved algorithms can be explored for better results.

## 3.2 MMI-Unscrambler: Multi-Modal Information Unscrambler

We introduce a Multi-Modal Information Unscrambler (MMI-Unscrambler) architecture to solve the problem of local inversion described above. MMI-Unscrambler is specifically designed to efficiently ingest multi-modal data that is present in the seismic traces and successfully disentangle subsurface information from traces. An overview of MMI-Unscrambler is shown in Figure 3a. MMI-Unscrambler first obtains embeddings for traces, combines them to get embedding of the context, which is further used for prediction of property model chunk. In what follows, we describe the architectural choices of MMI-Unscrambler and the rationale behind them.

### 3.2.1 Trace Embedding: Processing Multi-modal Information by Projection-concatenation

Multi-modal nature of seismic data arises due to different types of data associated with a trace. For example, a trace has (1) a time series amplitude information $\mathcal{D}(t)$ (2) source and receiver geo-location data $\mathcal{A}(t)$ (3) signature of disturbance created by air guns (time-series) (4) time of recording (time) (5) instrument sensitivity (float vector), etc. Ingestion of these data in a way that information obtained from them complement each other is important. MMI-Unscrambler achieves this by projection-concatenation. Let the different components of data associated with a trace t be denoted as $\{t[i]\}_{i=1}^{n}$. We first project each of the component via a separate function blocks to get individual embeddings and then concatenate the embeddings to get a final embedding of the trace. Formally,

$$\mathcal{E}(t) = [\mathcal{F}_1(t[1]), \mathcal{F}_2(t[2]), \cdots, \mathcal{F}_i(t[i]), \cdots]$$

Specifically in our case of traces, we have the following,

$$\mathcal{E}(t) = [\mathcal{F}_{aq}(\mathcal{A}(t)), \mathcal{F}_d(\mathcal{D}(t)), \ldots]$$

**MLPs for each $\mathcal{F}_i$s:** We propose to use MLP for all of these projections. The rationale for choosing MLPs is as follows: (1) [$\mathcal{D}(t)$, the signature of disturbance and other time-series data]: Recurrent Neural Networks (RNN) [32] and 1D-CNNs are popular architectures to process time-series data. Time-series data in SI has fixed frequency and number of recordings across different traces. Thus, RNN is not required to process these types of data. CNN, on the other hand, needs enough depth to capture long-range dependencies. In SI, the long-range dependencies are equally important. We propose to use MLPs that are efficient to compute and capture long-range dependencies in sequence gracefully. (2) [$\mathcal{A}(t)$, recording time, instrument sensitivity, etc] : MLPs are natural solutions for processing these types of data.

### 3.2.2 Context Embedding: Information Enhancement via Projection-mean Pooling

Embedding for context, $\mathcal{C}(q, w)$, is obtained by combining information from its traces, $\{t_i\}_{i=1}^{n}$. However, combining this information is non-trivial for the following reason. Each trace data is a superimposition of waves reflected from all over the subsurface. Hence, each trace captures relevant information about the entire subsurface. In this respect, raw data is a collection of highly redundant and correlated traces containing overlapping pieces of information. MMI-Unscrambler has to align the information of these pieces before combining them to predict property model block. We achieve the alignment via projecting the embeddings via functional block $\mathcal{G}$. We then enhance the information by mean pooling over all the projected embeddings.

Generally, sum-pooling is standard in NLP or embedding-based models. However,in a real survey (Figure 2), the number of traces in a context $\mathcal{C}(q, w)$ change with changing location $q$ of the context. It can be as low as tens of traces in context to as high as thousands of traces. This data characteristic

| Datasets | SEGSalt | | | RS-SEGSalt | | | GulfM-10 | GulfM-20 |
|---|---|---|---|---|---|---|---|---|
| Models | U-Net | Set-Transformer | MMIU | U-Net | Set-Transformer | MMIU | MMIU | MMIU |
| $\mathcal{L}_1$ | 196.82 | 232.78 | **163.77** | - | 294.76 | **162.79** | 135.27 | 164.99 |
| PSNR | 14.64 | 14.21 | **16.65** | - | 14.90 | **16.68** | 32.34 | 26.95 |
| SSIM | 0.45 | 0.40 | **0.47** | - | 0.34 | **0.41** | 0.82 | 0.75 |

Table 1: Summary of results of MMI-Unscramblerand baselines on all the four datasets. We use MMIU to denote MMI-Unscrambler.

prohibits us from using sum-pooling, which is common in NLP, where the number of words in context doesn't vary drastically. In our experiments, we observe that models trained with sum-pooling perform poorly than mean-pooling, especially when presented with extreme data with a very high or low number of traces per context block. The combined embedding can be viewed as an embedding for the context $\mathcal{C}$(q,w). Formally,

$$\mathcal{E}(\mathcal{C}(q, w)) = \frac{1}{n}\Sigma_{i=1}^{n}\mathcal{G}(\mathcal{E}(t_i))$$

### 3.2.3 Downstream Tasks

Once a context representation is obtained, we can now use this to predict any property we desire, such as velocity model, reflectivity model or even elastic properties of the subsurface. The property model is predicted by processing the context embedding using a function $\mathcal{H}$.

$$\hat{\mathcal{P}}(q, w_p, d) = \mathcal{H}(\mathcal{E}(\mathcal{C}(q, w)))$$

Again, we implement $\mathcal{H}$ as an MLP. Thus MMI-Unscrambler is essentially composed of all-MLP components, and we learn all these functional blocks in an end-to-end fashion. An overview of MMI-Unscrambler is shown in Figure 3a

### 3.2.4 Training Data

$(\mathcal{C}(q, w), \mathcal{P}(q, w_p, d))$ is a training data sample for training a mapping $\mathcal{M}$ of depth corresponding to $q$. Given a huge survey area, we can sample different points $q$ in the subsurface to generate multiple data points. Sometimes the availability of labeled real data is scarce. In such cases, we sub-sample the traces in context to generate more distinct training data points. Specifically, if $(\mathcal{C}(q, w), \mathcal{P}(q, w_p, d))$ is a data point for training, then so is $(c(q, w), \mathcal{P})$ for every non-empty $c(q, w)$ in power set of $\mathcal{C}(q, w)$. So, if we have ten traces in our context for a particular training sample, we can essentially have $2^{10} - 1 = 1023$ different training samples corresponding to this one input-output training sample. We also observe that generalization of the model is better when trained with augmented subsampled data. Because MMI-Unscrambler is robust to the different number of traces in the context, we can use subsampling to improve the model's inference time in SI.

### 3.3 Benefits of MMI-Unscrambler

**Robustness:** MMI-Unscrambler alleviates the issue of highly irregular spatial acquisition that plaques the convolution-based approaches.

**Ingestion of Multi-modal data:** Unlike current DL-based SI approaches, it scales easily to 5-dimensional real data. MMI-Unscrambler, with its projection-concatenation can ingest even higher dimensional seismic data obtained by adding varied available meta-data

**Resistance to overfitting:** Labeled data in seismic tasks can be scarce. However, for MMI-Unscrambler, we create exponentially more distinct training samples from a small sample in data. (see 3.2.4). Training model using this sampling approach prevents the model from overfitting. For example, SEGSalt data has 130 training samples, but the MMI-Unscrambler model with sampling technique does not overfit even when we run the model for over 1200 epochs. (see Figure 4)

## 4 Application and Empirical Results

We evaluate the MMI-Unscrambler against the competing methods on the velocity inversion problem on various datasets.

**Datasets:** To extensively evaluate MMI-Unscrambler against competing approaches, we select the following datasets.

- **SEGSalt:** (small, regular, 2D Velocity). SEGSalt is a small-scale, 2D velocity prediction dataset with the regular acquisition (Fig.6b). SEGSalt is an ideal dataset for the CNN-based U-net model used in [11] and is publicly available with an MIT license. We choose this data set to compare

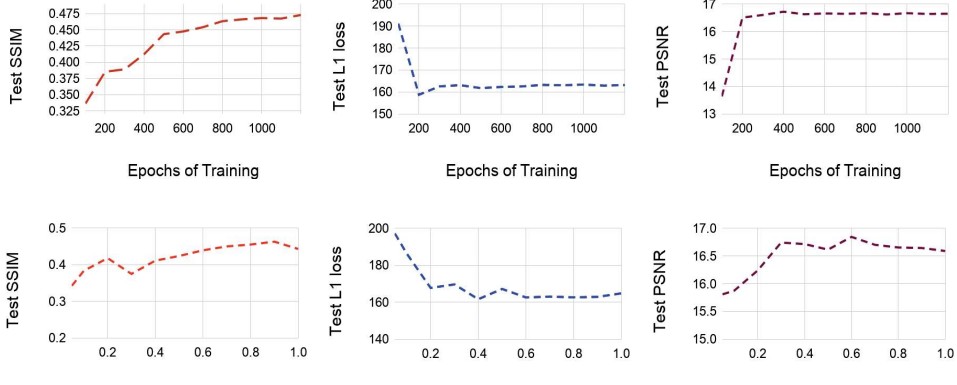

Figure 4: **Above:** Evolution of metrics SSIM, $\mathcal{L}_1$ loss and PSNR on test SEGSalt dataset as the training progresses to 1200 epochs. As can be seen SSIM always improves and $\mathcal{L}_1$ loss , PSNR achieve stable values. For elaboration on metrics see Experiments section. Model is trained with 0.8 fraction of traces. **Below:** Comparison on test performance when using different fractions of traces

against U-net at its best. **Data generation:** This data is generated by designing 2D velocity models and then running forward wave propagation using Devito [33][34] to obtain seismic traces. The acquisition geometry is regular: distributing 29 sources and 301 receivers uniformly along the surface line, with each receiver listening to each source. Details of data generation and other details are mentioned in supplementary B.1.1

- **RS-SEGSalt ( Realistically-sampled SEGSalt):** (small, irregular, 2D Velocity) RS-SEGSalt is obtained from SEGSalt by tweaking sampling the acquisition to make it more realistic. The way to sub-sample the traces is shown in Figure 6b. With such a simple tweak towards real data, the U-Net model breaks and cannot be applied. **Data generation**: For each sample in SEGSalt for each epoch, we randomly sample a fraction of contiguous traces. For experiments, we use a fraction uniformly sampled between $(0.7 \pm 0.25)$ and then choose the contiguous strips of receivers at a uniformly random location.

- **GulfM-10 and GulfM-20**( real, enormous, highly irregular, 3D Velocity) These are datasets we generate from large-scale real proprietary data obtained from a Big-Oil company. This dataset's acquisition geometry is highly irregular, as shown in Figure 2. The data scale is also immense ( (100Km x 100Km) survey area). Due to the 5-D trace data, large scale, and irregular geometry, this data set cannot be used with U-Net. We test set-transformer and MMI-Unscrambler on the auxiliary task of contextual velocity block prediction. We also show big block predictions obtained by applying a sequence of contextual predictions and stitching together an image. **Data generation:** For the task of contextual prediction, we sample data points as described in 3.2.4. The difference between GulfM-10 and GulfM-20 is the height of the target velocity block. GulfM-10 and GulfM-20 have velocity blocks with height 10% (1.5Km) and 20%(3Km) of the total velocity model, respectively. The block width and context width in both cases are 1Km and 5Km. The exact construction of the dataset and other details are described in supplementary B.1.2

**Baseline models:** We use two baselines for MMI-Unscrambler. (1) **U-Net:** [11] is the state-of-the-art model in recent literature for data-driven seismic processing. Also, it is the only model that was evaluated on publicly available SEGSalt data [2]. Therefore, we choose this open-source [11] implementation as the representative for U-Net. (2) **Set-Transformer:** [35] is the state-of-the-art model that introduce multi-head attention mechanisms on set representation. It has demonstrated its superiority in point cloud classification and anomaly detection. The hyper-parameters of these models can be found in supplementary B.2

**MMI-Unscrambler model:** The exact MMI-architecture details for the datasets described below are discussed in supplementary B.2. Briefly, we use the same number of layers for SEGSalt and GulfM-10/GulfM-20 datasets. The only difference is that we use a hidden layer of size 8192 in GulfM whereas 4096 in SEGSalt. Also, the input and output sizes of the network are determined by the dataset. More details on the architecture parameters is mentioned in B.2

---

[2]https://wiki.seg.org/wiki/SEG

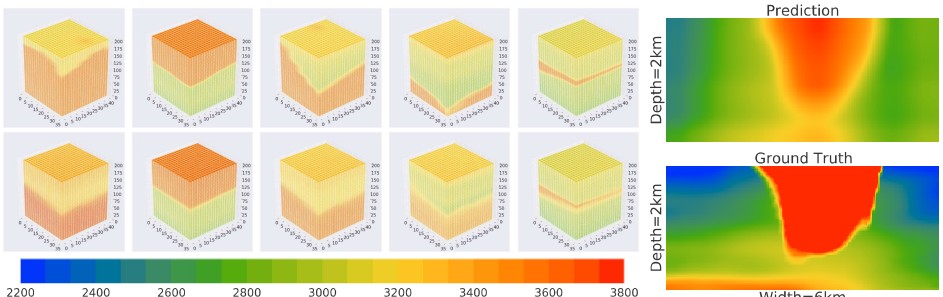

Figure 5: Visualization of prediction on GulfM-20 Dataset. **Left:** Top: ground-truth velocity cube, Bottom: predicted velocities. **Right:** cross-section visualization of inference on whole area. We want to stress that generating the above image from raw data via traditional methods takes several months because of the complex large scale data

**Metrics:** To evaluate the models, we use the following three metrics widely used in recent work on data-driven seismic processing [11, 13, 14, 36]. Namely, $\ell_1$ Loss (lower is better) , PSNR [37](higher is better) and SSIM [38](higher is better). $l1$ loss is $l_1$ metric applied to (input,output) images. SSIM is a structural similarity score widely used to measure the quality of images. PSNR (Peak Signal to noise ratio) is used to measure image reconstruction quality.

**Results:** We provide results in various forms:

- **Performance on metrics (Table 1)** : lists the performance of all the models on all the datasets on all metrics. The variance of $\ell_1$, PSNR and SSIM for MMI-Unscrambler in GulfM-10 dataset are 1.65, 0.92 and 0.02 respectively while in the same order are are 2.33, 1.07 and 0.01 in GulfM-20 dataset.
- **Metric evolution and robustness ( Fig. 4)**: top panel shows the evolution of metrics. The bottom panel shows the performance of the MMI-Unscrambler model trained on RS-SEGSalt when used for inference on different sizes of the sampled data.
- **Visualization (Fig. 5, 6a )**. Fig. 6a Shows the visualization of different models on SEGSalt. The test data visualization is presented in the supplement. We also show the performance of the MMI-Unscrambler model trained on RS-SEGSalt data and inferred on SEGSalt. In Fig. 5, we show visualisation on contextual block prediction task (left). Also, we run inference on larger blocks of velocity and show visualization (right). Details of visualization are presented in the appendix.

**Key highlights:**

- **U-Net vs. MMI-Unscrambler on SEGSalt:** On SEGSalt, which is the ideal data set for CNN based U-Net, MMI-Unscrambler performs better than U-net on all the metrics with significant margin (Table 1). This begs the question of whether convolutions are apt for seismic inversion. Convolution-based networks are known for their strong priors and disregard for long-range dependencies. MLPs, on the other hand, can capture all range dependencies and hence perform better. Visualizations in Figure 6a also echo similar results. At times, U-Net incorrectly predicts background velocity and shows weird salt structures. Though MMI-Unscrambler's output suffers from over-smoothening the boundaries, it captures detailed the salt shape and the background.
- **RS-SEGSalt vs. SEGSalt** Training the model on RS-SEGSalt leads to robust models due to the availability of extensive, varied data created by subsampling. Note that RS-SEGSalt only differs from SEGSalt in the input. Thus, we can train the model on RS-SEGSalt and test it on SEGSalt. We can see from Figure 6a that MMI-Unscrambler gives better salt shapes when trained on RS-SEGSalt. This is also reflected by better $\mathcal{L}_1$ and PSNR for MMI-Unscrambler.
- **MMI-Unscrambler vs Set-Transformer** Table 1 also demonstrates the superiority of MMI-Unscrambler over set-transformer. On regular observation, MMI-Unscrambler achieves over 20% performance boost on set-transformer. One intuition behind these results is that there is less dependency between traces. Thus, attending one trace with others would not improve the generalization of the set model.
- **MMI-Unscrambler on GulfM**. Figure 5 (right), shows that we get geologically sound predictions of the velocity model by using MMI-Unscrambler model. The image is obtained by stitching together the individual blocks predicted by MMI-Unscrambler model. Figure 5 (left) shows a mixture of small block predictions. SSIM on real velocity prediction is higher than that on synthetic data. Real velocities in a small block are relatively smooth, whereas synthetic data is filled with complexities to resemble the entire survey.

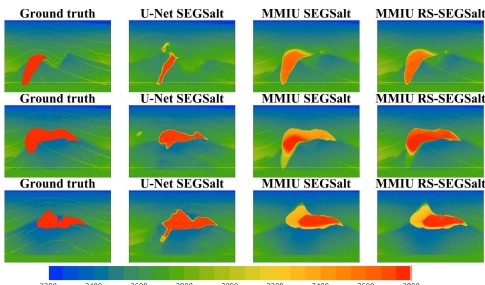

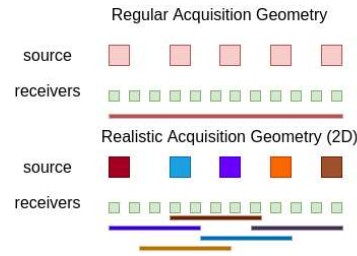

(a) Visualization of Prediction on SEGSalt Dataset : The prediction results for U-net vs MMI-Unscrambler model. MMI-Unscrambler performs particularly well when trained on RS-SEGSalt data. Also, MMI-Unscrambler model is robust to irregularities in the acquisition.

(b) Regular AG used in synthetic datasets is to have sources and receivers uniformly distributed on surface and have every receiver listen to every source. Realistic AG for 2D is designed to mimic Real acquisition. Here, randomly sampled contiguous segments of receivers are made to listen to each source. The color shows the pairings of receivers and sources.

Figure 6

**Experimental Hardware and Computation Power:** All experiments were performed on Tesla V100 GPUs. Training large models for gulf-of-Mexico datasets require around 10 hours of training.

## 5    Negative Societal Impact

This paper proposes an efficient seismic data processing approach. Seismic data processing is a fundamental problem in geophysics. It also has tremendous applications in earthquake estimation [39], ocean temperature profiling [40], and groundwater monitoring [41]. We hope that our MMI-Unscrambler could be extended to solve environmental protection problems. Moreover, MMI-Unscrambler reduces the expensive computation in seismic data processing by learning an end-to-end model to generate seismic velocity maps, etc. We believe this reduction would also achieve lower power consumption compared to the traditional seismic processing workflow. Furthermore, our method provides an accurate estimation of the subsurface velocity model, potentially preventing unnecessary exploration for the oil and gas company. We hope our effort will help reduce the negative effects of the current exploration system.

## 6    Conclusion and Future Work

To tackle the scale of seismic inversion, we introduce Aux-SI and define the auxiliary task of contextual small property model block prediction to solve the problem of large-scale property model (in Gigabytes) prediction from Terabytes of data. We can efficiently train models for this auxiliary task. We alleviate the issues of multi-modal data and highly irregular geometry of acquisition by using a novel MMI-Unscrambler architecture to align and combine available data. Our model achieves geologically sound predictions on real data at its original scale. We believe this is the first-ever demonstration of a model on real-scale seismic data. In the future, we plan to further improve information utilization by including all metadata available. Also, we would explore better stitching algorithms to perform large-scale inference. Currently, we deploy a supervised approach to SI. This requires the availability of pre-computed velocity models, which is difficult to obtain. Exploring unsupervised DL-based techniques is also an interesting future direction.

## Acknowledgments

This work is supported by a research grant from Shell, National Science Foundation IIS-1652131, BIGDATA-1838177, AFOSR-YIP FA9550-18-1-0152, ONR DURIP Grant, and the ONR BRC grant on Randomized Numerical Linear Algebra.

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
