# Appendix

## A  Notation

| Notation | Meaning |
|---|---|
| $t$ | trace |
| $q \in R^3$ | location |
| $u(s,q) \in R$ | amplitude |
| $\mathcal{A}(t)$ | acquistion geometry of trace |
| $\mathcal{D}(t)$ | time series data of trace |
| $\mathbb{A}$ | complete acquistion geometry |
| $\mathcal{P}$ | complete property model. |
| $w$ | width of blocks |
| $d$ | depth of blocks |
| $\mathcal{P}(q,w,d)$ | property model chunk |
| $t.data_i$ | a particular type of data associated with trace t. $\mathcal{A}(t), \mathcal{D}(t)$ are examples. |
| $\mathcal{C}(q,w)$ | Context at q of size w |
| $Area(q,w)$ | 2d area around q of size w |
| $\mathcal{M}$ | model |
| $\mathcal{M}(\mathbb{d})$ | model at a depth |
| $\mathcal{F}$ | functional block |

Table 2: summary of notation

## B  Experiment Details

In this section, we elaborate on the various details of our experiments. First we talk about the data generation for simulation and real dataset. Then, we talk about the MMI-architectures used in both cases. The exact MMI-model architecture when applied to velocity inversion can be seen as in Figure 7

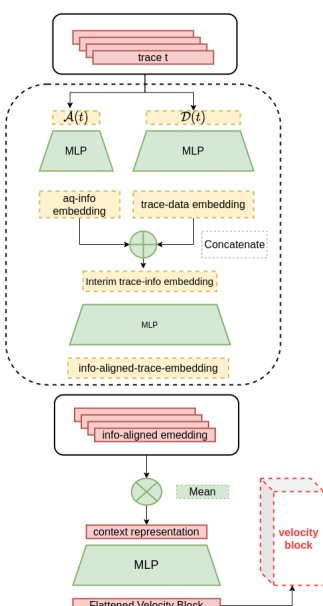

Figure 7: MMI-Unscramblerapplied to velocity inversion

### B.1 Data generation for training

### B.1.1 Seismic Data Simulation

The 2D SEG salt models in SEGSalt dataset [11] are identical to each other. The data statistics are shown in Table 3. [11] assume that the velocities varies from 2000 m/s to 4500 m/s, where the material with 4500 m/s is the salt body [1]. The shape of the target velocity matrix is $201\times301$ with a spatial interval of 10m.

The simulation parameters for trace generation are shown in Table 4. We perform a the time-domain stagger-grid finite-difference scheme to simulate traces by acoustic wave equation. The scheme contains both a second-order time direction and eight-order space direction [42]. For acquisition geometry, we use 301 receivers that are uniformly distributed at a constant spatial interval and 29 sources. All recievers listen to all sources. There also exists a perfectly matched layer (PML) [43] absorbing boundary condition that reduces unphysical reflection on edges. The trace is 2000 dimensional vector and each velocity matrix corresponds to $29 \times 301$ traces. To fit the requirements of the CNN, each trace is down-sampled to 400 dimension by [11]. Then, the 301 traces of each shot formulate a $301 \times 400$ matrix. Therefore, the U-Net operations can performed on $301 \times 400 \times 29$ image. Our MMI-Unscrambler model formulate each sample as a set with size $n < 29 \times 301$ and element dimensionality $400$. We sub-sample the $29 \times 301$ traces to generate set embedding and predict the target velocity matrix.

| Dataset | Training Samples | Testing Samples | Velocity Shape |
|---------|------------------|-----------------|----------------|
| SEGSalt | 130 | 10 | $201\times301$ |

Table 3: Data Statistics for VMB

| Number of Source | Number of Receivers | Sampling Frequency | Ricker Wave | Simulation Time | Traces Length |
|------------------|---------------------|--------------------|-------------|-----------------|---------------|
| 29 | 301 | 1kHz | 25Hz | 2s | 2000 |

Table 4: Simulation Parameters

### B.1.2 Training data for Gulf-M datasets

The GulfM dataset is generated from the real marine survey data shared by one of the major energy and petrochemical company. As shown in the name, the raw seismic data is obtained by oil and gas platform [44] in the gulf of mexico. We were given a single velocity image of the sub-surface till depth approximately 10km and area 100Km x 100Km. Each trace has 7008 recordings. The large ground-truth 3D velocity models are generated by geophysicists with duration over 8 months. In the GulfM datasets, the resolutions are 50m/pixel in X, 60m/pixel in Y, and 32m/pixel in Z. These resolutions are standard and determined by the exploration team in the big oil company. In order to generate training and testing data from this large image, we first divide the data into two regions, the training samples are drawn from first region and testing samples are drawn from another region. As mentioned in the paper, we should build different models at different depths. In this particular case we focus at a depth of 3Km from the surface of water.

**GulfM-20 dataset** In this dataset, the target velocity block is of size $17 \times 20 \times 100$ pixels. i.e. $1Km \times 1Km \times 3Km$. The relevant context area is of size 5Km x 5Km around this target block. We sample this target block randomly in the training area ( or testing area). The traces obtained in this context is the input to our model. The number of traces in the context varies from 2 to 8779.

**GulfM-10 dataset** The output velocity block is $17 \times 20 \times 50$ pixels. That is, 1km $\times$ 1Km $\times$ 1.5 Km. The input again is the context of 5Km $\times$ 5 km around the target velocity block.

We did not perform any hyperparameter tuning on block size and context size prediction. These sizes were chosen as reasonable sizes in consultation with the Geophysicists at the big oil company.

**Data statistics** The data statistics are mentioned in the table below. The number of samples 400 are further subsampled to generate exponential number of examples as per sampling methodology described in paper.

| Dataset | Training Samples | Testing Samples | Velocity Shape |
|---|---|---|---|
| GulfM-10 | 400 | 100 | $17\times20\times50$ |
| GulfM-20 | 400 | 100 | $17\times20\times100$ |

Table 5: Data Statistics for VMB

## B.2 Network Architecture

The network architectures used for segsalt data is mentioned in Table 6 and those for GulfM-datasets is mentiond in Table 7. The total number of parameters for GulfM-10 is 700M and that for GulfM-20 is 840M. For set transformer [35], we implement an encoder with a stack of set attention blocks. Then, we introduce a pooling by multihead attention module. Finally, we have a stack of set attention blocks in decoder. We set the dimensionality of all hidden layers to 4096 and the number of attention heads to 4.

| Functional block | Arch | Specifics | Activation | Parameters |
|---|---|---|---|---|
| Trace Embedding | 4-layer MLP | 400-10240-4096-4096-4096 | ReLU | 79M |
| Location Embedding | 4-layer MLP | 4-512-512-512-512 | ReLU | 0.79M |
| Combined Trace Embedding | 2-layer MLP | (4096+512)-4096-4096 | ReLU | 35M |
| Velocity Model Prediction - Hidden | 2-layer MLP | 4096-4096-4096 | ReLU | 33M |
| Velocity Model Prediction - Output | Linear | 4096-60501 | - | 240M |

Table 6: Architecture used for simulation data.

| Functional block | Arch | Specifics | Activation | Parameters |
|---|---|---|---|---|
| Trace Embedding | 4-layer MLP | 7008-10240-8192-8192-8192 | ReLU | 289M |
| Location Embedding | 4-layer MLP | 4-512-512-512-512 | ReLU | 0.79M |
| Combined Trace Embedding | 2-layer MLP | (8192+512)-8192-8192 | ReLU | 138M |
| Velocity Model Prediction - Hidden | 2-layer MLP | 8192-8192-8192 | ReLU | 134M |
| Velocity Model Prediction - output | Linear | 8192-(17000 or 34000) | - | 140M / 280M |

Table 7: Architecture used for real data.

**hyper-parameter tuning**

(1) Model architecture parameters: The good news is that we can generate as many informative samples as we need ( mentioned in section 3.2.4 on line 232 ). With abundant training data, we just had to ensure that the model did not underfit. As a result, we didn't do much hyperparameter tuning.

(2) Learning rate: we tried different learning rates and learning rate schedules. Finally, we decided to start with a small learning rate $(10^{-5})$ and a learning rate scheduler to reduce the learning rate every 25 epochs. The total training epochs are 150 epochs.

The same hyperparameters were used across datasets and worked well.

## B.3 Model Training

We train both MMI-Unscrambler model and U-Net on a server with 1 Nvidia Tesla V100 GPU and two 20-core/40-thread processors (Intel Xeon(R) E5-2698 v4 2.20GHz). MMI-Unscrambler, set transformer and U-Net use Adam as optimizer. The training hyper-parameters for MMI-Unscrambler, U-Net and set transformer follow the paper [11] with learning rate modifications to make better performance. The variance of $\ell_1$, PSNR and SSIM for MMI-Unscrambler in GulfM-10 are 1.65, 0.92 and 0.02. The variance of $\ell_1$, PSNR and SSIM for MMI-Unscrambler in GulfM-20 are 2.33, 1.07 and 0.01.

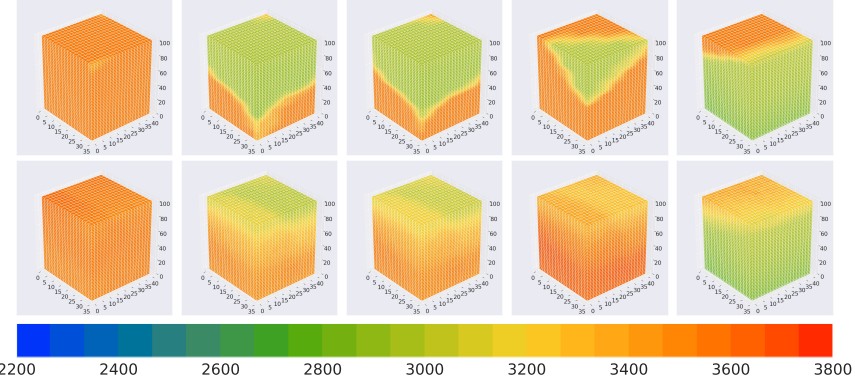

Figure 8: Visualization of prediction on GulfM-10 dataset

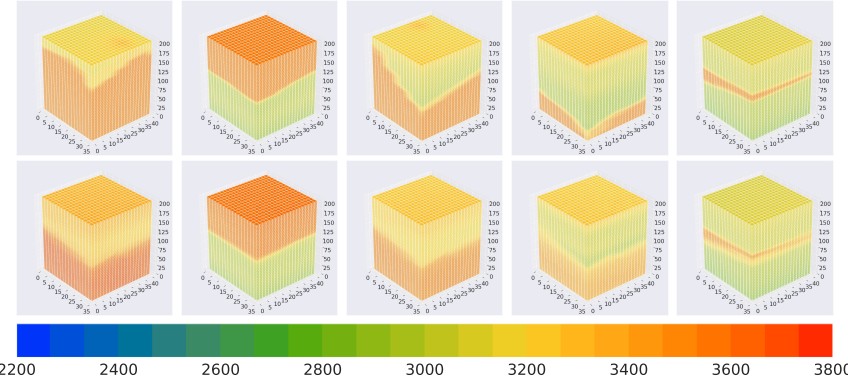

Figure 9: Visualization of prediction on GulfM-20 dataset

## B.4 Visualizations

We present the visualization of all predictions in SEGSalt dataset in Figure 10. A sample prediction for GulfM-10 and GulfM-20 dataset is shown in Figure 8 and Figure 9. Here, we choose different color maps for demonstrate the details information in seismic velocities. For the inference on large scale in Figure 11, we run the model in the region where both GulfM10 and GulfM20 datasets are randomly sample from.

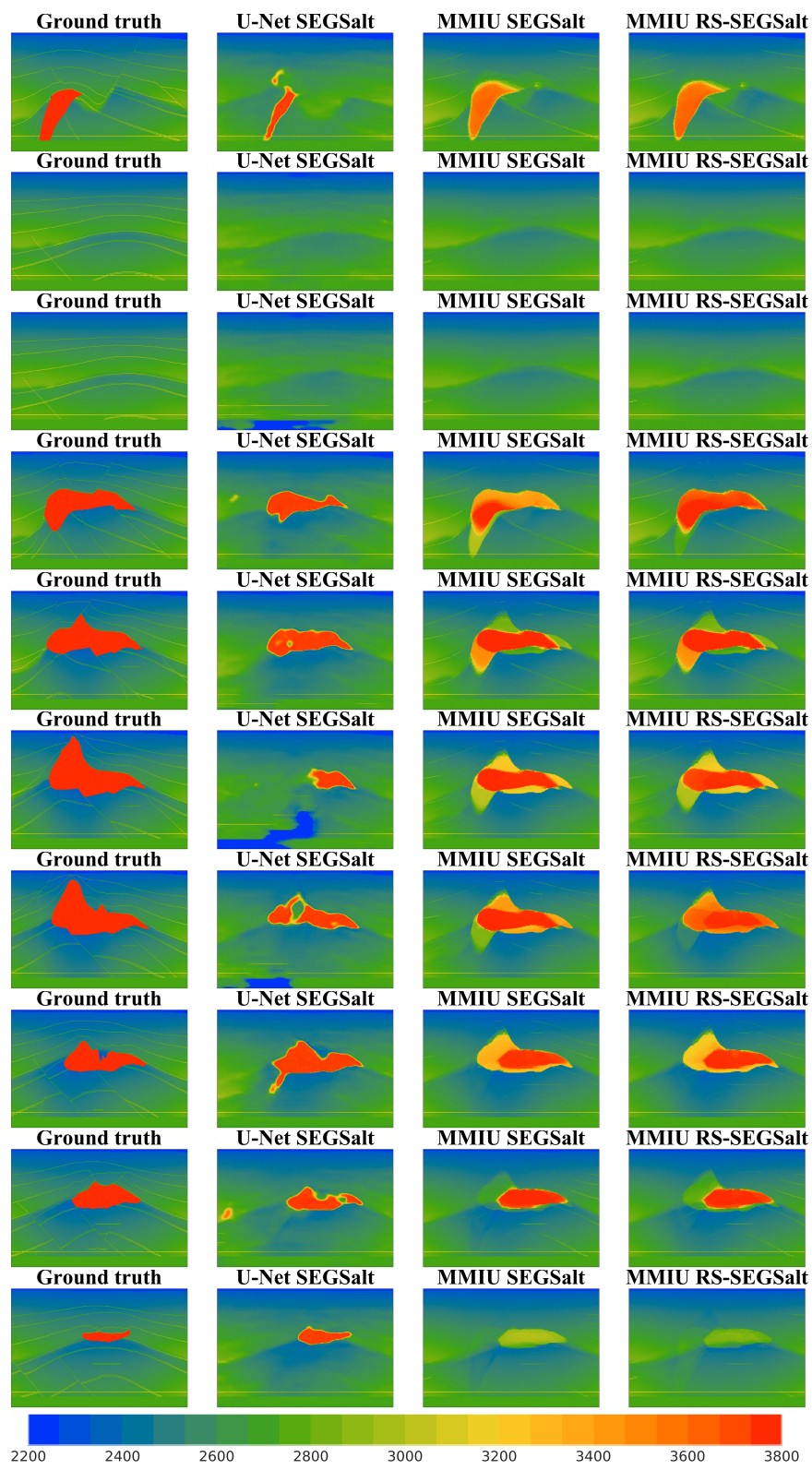

Figure 10: Visualization of prediction on SEGSalt dataset. We denote MMI-Unscrambler as MMIU for simplicity.

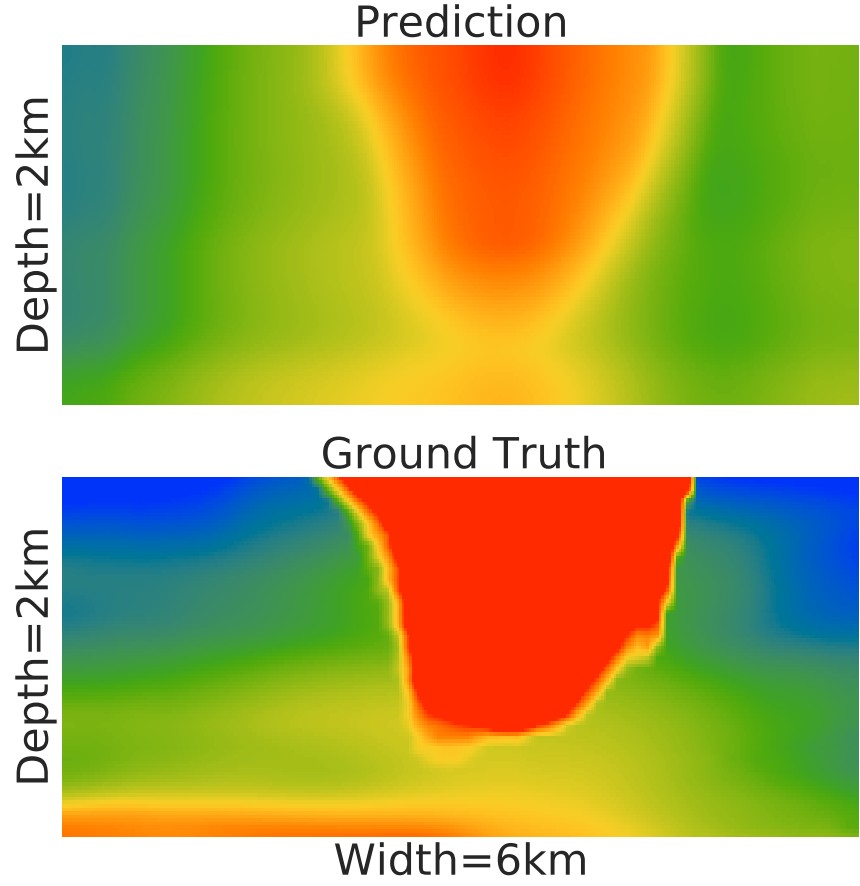

Figure 11: Visualization of prediction on large scale VMB in a depth 2km and width 6km area.