# OpenReview forum: "Raw Nav-merge Seismic Data to Subsurface Properties with MLP based Multi-Modal Information Unscrambler"
_NeurIPS.cc/2021/Conference — NeurIPS 2021 Poster_

### Official Review · Reviewer_ahiS · 2021-07-08

**Rating:** 7
**Confidence:** 4

**Summary:**

This study proposes a new method to the problem of seismic inversion, which is an ill-posed inverse problem of major importance for understanding the composition of the earth subsurface that is used in many applications. The proposed method relies on simple MLP blocks, but organized in a clever way in order to be able to use multi-modal, size-varying and sparse input data. Moreover, the separation in different spatial blocks allows the model to be efficient and parallelizable. The results are performed on different data sets, and compared with more standard time-series or spatial methods, such as Unet and Transformer-based methods. The proposed approach seems promising and robust, which is what is needed for being applied in the real world.

**Ethical Concerns:**

Could be, see section above.

**Ethics Review Area:**

["Inappropriate Potential Applications & Impact  (e.g., human rights concerns)"]

**Limitations And Societal Impact:**

This paper is meant to be used to find Oil for Big Oil companies. So it is subject to personal opinion whether this application is positive or negative in terms of societal impact. Yet, it was clear in the paper from the start what is the goal, it was not hidden, and I personally think that since we, authors and reviewers, are all currently relying on Oil and gaz to live our lives, being able to find it easily is a positive impact. I hope that the company that payed for this work also spends efforts in developing other energy or even better, tries to find ways to reduce energy consumption.
Yet, the authors could remind in the societal section a sentence regarding the final application of the work.

**Main Review:**

The paper is interesting and worth publishing. Yet, I found that the paper was not sufficiently proofread and it was difficult to follow. I hope the authors can spend time in correcting all the typos, in changing the variable names that are not clear, and adding the missing details. Specifically, please answer the following questions:
1) when you talk about 5D data, do you mean 3D+time (so 4D), and what is the last dimension?
2) Please give more details on your different MLP architectures: for now we don't have any information about their size, their number of layers, their activation functions, etc.
3) what is the size of the bock and of the context that you used for the different data sets, and more importantly how did you chose it? I only saw the size in one of the experiments.
4) how did you perform the hyperparameter tuning? what parameter were finally selected (size of blocks, learning rate, size of embeddings..)? was the same values used for all the different data sets or not?
5) how many samples did you finally used in practice for your training, even if there was overlap between samples thanks to data augmentation?
6) explain better how you divide your train validation and test samples. Since your data augmentation technique has a large overlap between samples, I just want to check that the separation between the sets is done correctly.
7) please add the 3rd part of your pipeline, the 'downstream tasks', in your figure 3.a. Add also a proper legend so we can understand the figure on its own.
8) CNN models are able to capture long range dependency... that is why there are multiple layers with max poolings, so the dependencies at different scales can be captured. Please explain better why it might not be relevant for this application.
9) Would you say that what actually makes the difference in your method is to separate the problem in small blocks? Could you perform the same division using the other state-of-the-art methods?
10) explain more clearly what is G (I guess it's an MLP, from the figure, but it is never said in the main text).
11) 'q.z' formulation is not clear, I guess you meant the vertical component of the position q? Same, there is a confusion between the time 't' and the trace 't'. please use another name for the trace so we can understand the text and the figure. Last, in Figure 3b, explain what are the 3 axis of the cube.
12) would it be possible to interpolate the traces over the surface area in order to have a grid (or otherwise a graph of sensors)? why would that lead to larger errors than your method (if it does)?

typos: please check your paper again as it is not my job to proofread papers before the authors.
- problem deﬁnition 1can
- a separate function blocks
- 2 1 0 − 1 --> 2^⁽10)-1 (I guess??)
- Generally sum-pooling in standard in NLP or embedding based models. --> no verb
- many spaces missing
- many points missing
- check capital letters


**Needs Ethics Review:**

Yes

**Time Spent Reviewing:**

2

---

> ### Author Response · Authors · 2021-08-10
> **Thank you for the comprehensive review, please see the following clarifications.**
>
> Thank you for your support of the paper.  We answer the questions raised point wise:
>
> **1. when you talk about 5D data, do you mean 3D+time (so 4D), and what is the last dimension?**
>
> The 5D data arises from two dimensions of the source location ( source.x and source.y), two dimensions for receiver location (receiver.x, receiver.y), and one dimension for trace/time series. Interestingly MMI-Unscrambler can scale seamlessly to even more dimensional data obtained by adding more modalities of information.
>
>
> **2.  Please give more details on your different MLP architectures: for now we don't have any information about their size, their number of layers, their activation functions, etc.**
>
> The details of the model architecture are as follows: (a) we use a one-layer MLP to map each trace signal into 10240-dimensional embedding.  Then, we perform a three-layer MLP on the trace embeddings. Each MLP layer has 4096 hidden units.  (b) For each trace’s 4D location information, we use an MLP to map it into a 512 dimension vector. Then, we concatenate the trace embedding with the location embedding and perform a five-layer MLP with 4096 hidden units in each layer. Finally, we combine the trace information by taking their average and complete the final prediction via one layered MLP. For each MLP layer used in the paper, we perform ReLU activation afterward.  We will add more details on network architecture to the paper.
>
> **3. what is the size of the block and of the context that you used for the different data sets, and more importantly how did you choose it? I only saw the size in one of the experiments.**
>
> The output of MMI-Unscrambler is a 3D chunk of the velocity model. In seismic imaging, we denote the 3D dimension as X, Y and Z. X and Y are spatial dimensions on the surface and Z is the depth. In the GulfM datasets, the resolutions are  50m/pixel in X, 60m/pixel in Y, and 32m/pixel in Z. These resolutions are standard and determined by the exploration team in the big oil company.  In the GulfM-10 dataset, the output is a 17x20x50 3D voxel array where one voxel is 50m x 60m x 32m. The input, on the other hand, is all the traces that appear in the area of the 5Km x 5Km area around the output block. We did not perform any hyperparameter tuning on block size and context size prediction. These sizes were chosen as reasonable sizes in consultation with the Geophysicists at the big oil company.
>
>
> **4. how did you perform the hyperparameter tuning? what parameters were finally selected (size of blocks, learning rate, size of embeddings..)? was the same values used for all the different data sets or not?**
>
> (1) size of blocks: We did not perform any hyper-parameter tuning on the size of prediction blocks and context sizes. These sizes were chosen as reasonable sizes in consultation with the Geophysicists at the big oil company.
>
>
> (2) Model architecture parameters: The good news is that we can generate as many informative samples as we need ( mentioned in section 3.2.4 on line 232 ). With abundant training data, we just had to ensure that the model did not underfit. As a result, we didn't do much hyperparameter tuning.
>
> (3) Learning rate: we tried different learning rates and learning rate schedules. Finally, we decided to start with a small learning rate (10^-5) and a learning rate scheduler to reduce the learning rate every 25 epochs. The total training epochs are 150 epochs.
>
> The same hyperparameters were used across datasets and worked well.
>
> **5. how many samples did you finally used in practice for your training, even if there was overlap between samples thanks to data augmentation?**
>
> For the GulfM dataset, we perform training on a survey area with 400 training samples. We augment data while preparing a batch by following the procedure mentioned in section 3.2.4 on line 232. This leads to a distinct (with high probability) batch for each iteration.   Then, we test our model on a different survey area with 100 training samples. We have a target 3D velocity cube as a label in each sample and the corresponding seismic wave signals as data in each sample.
>
> **6. explain better how you divide your train validation and test samples. Since your data augmentation technique has a large overlap between samples, I just want to check that the separation between the sets is done correctly.**
>
> We train and test on separate subsurface areas in the survey.  So the predictions are not seen during training.
>
> **7. please add the 3rd part of your pipeline, the 'downstream tasks', in your figure 3.a. Add also a proper legend so we can understand the figure on its own.**
>
> Thanks for the suggestion. We will improve the figure.
>
> **8.CNN models are able to capture long range dependency... that is why there are multiple layers with max poolings, so the dependencies at different scales can be captured. Please explain better why it might not be relevant for this application.**
>
> CNN would need to have significant depth to capture long-range dependencies. However, this is not the reason why CNN is not suitable for seismic data.  Applying CNN to the Aux-SI problem is known to become infeasible for many reasons. (1) Data is inherently irregular. (2) Filling missing data to make it regular creates a prohibitively large 5D data matrix. For instance, in our GulfM-10 experiment, the discretized context area is 85 x 100. Hence, the number of source-receiver configs is 8500^2. With an additional trace dimension of 7008, the total size of context would be around 500GB).
>
> **9. Would you say that what actually makes the difference in your method is to separate the problem in small blocks? Could you perform the same division using the other state-of-the-art methods?**
>
> We believe that breaking this complex problem in small block prediction (Aux-SI) was the main reason we could efficiently solve the velocity model prediction problem to a reasonable extent. The current state-of-the-art baselines are only for structured 2D simulation data and do not scale to unstructured real data. No state-of-the-art model for this particular problem is known as of yet. We successfully show that MMI-Unscrambler is an efficient model architecture for this problem.  Testing different popular architectures for the Aux-SI problem would be good future work.
>
>
> **10. explain more clearly what is G (I guess it's an MLP, from the figure, but it is never said in the main text).**
>
> $G$ is indeed the MLP. We will fix the text to make it clear.
>
> **11.'q.z' formulation is not clear, I guess you meant the vertical component of the position q? Same, there is a confusion between the time 't' and the trace 't'. please use another name for the trace so we can understand the text and the figure. Last, in Figure 3b, explain what are the 3 axis of the cube.**
>
> $q.z$ indeed means the depth of point $q$. We will make it clear in the text.  In Figure 3b, the 3 axes are spatial location x,y, and depth location z.
>
> **12.would it be possible to interpolate the traces over the surface area in order to have a grid (or otherwise a graph of sensors)? why would that lead to larger errors than your method (if it does)?**
>
> In the Segsalt dataset,  our baseline U-Net regards the traces and a grid and interpolate them as an image, we show that MMI-Unscrambler with irregular geometry (RS-Segsalt) performs better than CNN based U-Net.
>
> We thank the reviewer for identifying the typos. We will update the paper with a clear version.

---

> > ### Comment · Reviewer_ahiS · 2021-08-10
> > **Interesting answers, keeping my 7 rate**
> >
> > Thank you for answering my questions, I think your answers are very interesting and I hope they will be included in the paper as it will improve it.
> > I am keeping my 7 rate as the answers are coherent with my first opinion. I hope the paper will be accepted.

---

> > > ### Author Response · Authors · 2021-08-10
> > > **Thanks for the support.**
> > >
> > > We thank the reviewer for the support of our paper!

---

> > ### Comment · Reviewer_ahiS · 2021-08-10
> > **limitations and societal impact**
> >
> > While I wrote a paragraph on the societal impacts, you did not mention at all this in your response. Please provide some additional sentences to the societal impact paragraph of your manuscript that did not even mention the problem at first...

---

> > > ### Author Response · Authors · 2021-08-10
> > > **Societal Impact Paragraph**
> > >
> > > We thank the reviewers for the concern. We will add an additional section to discuss the social impact. Here we summarize the key points:
> > > 1. The seismic data processing is a fundamental problem in geophysics. It also has tremendous applications in earthquake estimation[1], ocean temperature profiling[2], and groundwater monitoring[3]. We also hope that our method could be extended to solve environmental protection problems.
> > >
> > > 2. Our method reduces the expensive computation in seismic data processing by learning an end-to-end model to generate seismic velocity maps etc. We believe this method would also reduce the power consumption of traditional seismic processing workflow.
> > >
> > > 3. Our method provides an accurate estimation of the subsurface velocity model, which could potentially prevent unnecessary exploration of the oil and gas company. We hope our effort will help reduce the negative effects of the current exploration system.
> > >
> > >
> > > [1] Bakun, W. U., & Wentworth, C. M. (1997). Estimating earthquake location and magnitude from seismic intensity data. Bulletin of the Seismological Society of America, 87(6), 1502-1521.
> > >
> > > [2] Wood, W. T., Holbrook, W. S., Sen, M. K., & Stoffa, P. L. (2008). Full waveform inversion of reflection seismic data for ocean temperature profiles. Geophysical Research Letters, 35(4).
> > >
> > > [3] Steeples, D. W., & Miller, R. D. (1988, January). Seismic reflection methods applied to engineering, environmental, and ground-water problems. In Symposium on the Application of Geophysics to Engineering and Environmental Problems 1988 (pp. 409-461). Society of Exploration Geophysicists.

---

### Official Review · Reviewer_XYVo · 2021-07-13

**Rating:** 7
**Confidence:** 4

**Summary:**

The paper tackles the scalability issue of seismic inversion in real-world subsurface and property estimation. The authors propose the Aux-SI framework, where Multi-Modal Information Unscrambler (MMI Unscrambler) is designed, which breaks the large problem into local inversion problems and then aggregates the multi-model information. Extensive experiments are conducted, on both synthetic and real massive datasets, to show the effectiveness of the proposed method.

**Main Review:**

This paper is well organized and presented. The main strength is the successful application of machine learning on large-scale, real-world problems. As the authors mentioned, deep learning models in such subsurface property estimation has not been applicable on the real industry level. This paper breaks the limit. Thus, I believe this work could be interesting to many NeurIPS audiences.

The problem studied is interesting to me, and using DL in this problem seems quite novel and new. The empirical results are comprehensive and convincing that the proposed method can do a good job in real-world scenarios. Thus, I think this is in general a good application paper that may interest the many practitioners, and may originate more future efforts in combining machine learning with geological prospecting or oil exploration.


Questions:

What is the input dimensionality to the first MLP layer? Is that possible to break the high dimensional input (eg, 5D) into lower dimensions and apply CNN like U-Net?

Could you explain more on why MLP is used instead of CNN and RNN, and the pattern of the time-series data in SI?

**Time Spent Reviewing:**

2

---

> ### Author Response · Authors · 2021-08-10
> **Thank you for the comprehensive review, please see the following clarifications.**
>
> We thank the reviewer for your support of the paper.  We answer the questions raised point wise :
>
> **1) What is the input dimensionality to the first MLP layer? Is that possible to break the high dimensional input (eg, 5D) into lower dimensions and apply CNN like U-Net?**
>
> (a) The first set of MLP layers ingest traces and location information (this can be extended to more modalities of data available). In GulfM datasets, the MLP that ingests traces has an input dimension of 7008, and the one that ingests location information has dimension 4.
>
> (b) The 5D, in the seismic world, refers more to the multi-modality of the information (spatial and temporal time series, variable and non-uniform locations of source and receiver) rather than the five components of a vector. The resulting dimensionality in terms of input size is huge. Our method is already doing several dimensionality reductions and alignment using the MMI-Unscrambler architecture.
>
> (c) Applying CNN to the Aux-SI problem is known to become infeasible for many reasons. (1) Data is inherently irregular. (2) Filling missing data to make it regular creates a prohibitively large 5D data matrix. For instance, in our GulfM-10 experiment, the discretized context area is 85 x 100. Hence, the number of source-receiver configs is 8500^2. With an additional trace dimension of 7008, the total size of context would be around 500GB).
>
>  We are excited that we can still tame this multi-modal complex data and can create models with statistically significant predictive power.
>
>
> **2)Could you explain more on why MLP is used instead of CNN and RNN, and the pattern of the time-series data in SI?**
>
>
> The seismic data is multi-modal, large, and complex. Most CNN and RNN go out of memory even with the most powerful GPUs.  This is one of the main reasons why we wanted to stay with MLP with the real dataset.  Also, the community is now converging on the prediction power of MLPs being equivalent to CNNs. Additionally, MLPs are known to be  efficient for inference and require much lower memory as compared to CNNs, which require extensive feature maps for high-resolution images。

---

> > ### Comment · Reviewer_XYVo · 2021-08-26
> > **Thanks for the response**
> >
> > Thanks for the response. The rebuttal well addressed my question. I will keep my score.

---

> > > ### Author Response · Authors · 2021-08-31
> > > **Thanks for the support**
> > >
> > > We thank the reviewer for their support for our paper.

---

### Official Review · Reviewer_2jDV · 2021-07-16

**Rating:** 6
**Confidence:** 3

**Summary:**

This is an interesting paper that proposes to leverage MLP for solving seismic imaging problems, which are well known to be data-intensive and heavy in computation.
The paper provides an easy-to-follow introduction of the problem of seismic imaging and clearly illustrates how data-driven methods can potentially reform the seismic processing workflow, showing great promise of this specific research direction.
The proposed method, *MMI-Unscrambler*, consists of three phases. In the first phase, every trace $t\in C(q,w)$ is mapped to embeddings by individual MLP for each different modal information $t[i]$. A concatenation operation is subsequently used to form one single embedding for each trace. In the second phase, the set of embeddings associated with $C(q,w)$ is further mapped by another MLP (shared across different traces) to align these embeddings followed by average pooling. The rationale for this is that each trace has a dramatically different geometric shape and can not be directly aligned.
In the last phase, a MLP maps the aligned embedding of $C(q,w)$ to the final value $P(q,w_p,d)$.

The authors extensively validate their method on three datasets of increasing difficulty. In particular, the last dataset is real, huge, and irregular. In all datasets, MMI-Unscrambler outperforms U-Net and Set-Transformer baselines with noticeable improvements in terms of numerical values, which supports their aforementioned claims.


**Limitations And Societal Impact:**

There are some confusions that need the authors' clarification:

1. On what resolution level is the MMI-Unscrambler trained? Is the input of MMI-Unscrumbler a point (voxel) in the $C(q,w)$ and the output a point in $P(q,w_p,d)$? or MMI-Unscrambler takes and outputs a chunk of data, like a slice?

2. Figure 3.(a) is not very clear. What is the meaning of the red circle? The notation $e_i$ is not defined. Maybe $C(q,w)$ is clearer than $C$ since in the text it is always $C(q,w)$. The caption is not informative. Please consider having a self-contain caption. In fact, Figure 7 in the supplement should be used as a replacement for Figure 3(a).

3. In Figure 3.(b), what do different colors mean at the same level (i.e. dark and light red)?

4. Maybe the supplement should be referenced more often in the main paper since it provides necessary information for understanding the technical details.

5. For reproducibility, the author should consider publishing their code. Also, a comparison of sum-pooling and average-pooling will better justify the choice of the latter.

6. The scalability of the network should be discussed. For example, how many MLPs are needed to cover the gulf dataset in the experiment?

**Main Review:**

**Originality:** I am not a fully active researcher in seismic imaging, but as far as I know, this paper first proposes an MLP-based network for this topic.

**Quality:** The experimental validation in this paper is comprehensive and convincing. The quality of this paper is high.

**Clarity:** The paper did a very good job in explaining the background of seismic imaging. Figure 1 clearly shows how data-driven methods will reform the overall reconstruction pipeline, helping draw more attention to seismic imaging or waved-based inversion. However, there is some unclarity in the notations that can make the paper not easy to follow.

**Significance:** The paper is significant because it extends the recent popular MLP-base learning to seismic imaging.

**Time Spent Reviewing:**

3 hours

---

> ### Author Response · Authors · 2021-08-10
> **Thank you for the comprehensive review, please see the following clarifications.**
>
> We appreciate your concise and precise summarization of our work! Following up on your suggestions, we will add more details in the main paper (or supplementary due to the space limits) to clarify areas of concern.
>
> Below we provide more clarification on the details:
>
> **1)  On what resolution level is the MMI-Unscrambler trained? Is the input of MMI-Unscrumbler a point (voxel) in the $C(q,w)$ and the output a point in $P(q,w_p,d)$? or MMI-Unscrambler takes and outputs a chunk of data, like a slice?**
>
> The output of MMI-Unscrambler is a 3D chunk of the velocity model. In seismic imaging, we denote the 3D dimension as X, Y and Z. X and Y are spatial dimensions on the surface and Z is the depth. In the GulfM datasets, the resolutions are  50m/pixel in X, 60m/pixel in Y, and 32m/pixel in Z. These resolutions are standard and determined by the exploration team in the big oil company.  In the GulfM-10 dataset, the output is a 17x20x50 3D voxel array where one voxel is 50m x 60m x 32m. The input, on the other hand, is all the traces that appear in the area of the 5Km x 5Km area around the output block.
>
>
> **2)Figure 3.(a) is not very clear. What is the meaning of the red circle? The notation $e_i$  is not defined. Maybe $C(q,w)$ is clearer than $C$  since in the text it is always $C(q,w)$. The caption is not informative. Please consider having a self-contain caption. In fact, Figure 7 in the supplement should be used as a replacement for Figure 3(a).**
>
>  Red circle implies "mean" operation performed over embeddings obtained from the previous layer. We will fix the figure incorporating all the comments.
>
> **3)In Figure 3.(b), what do different colors mean at the same level (i.e. dark and light red)?**
>
> The different colors along with arrows attempt to show the prediction of one block at a time while inferring the entire subsurface. We will improve our figure to make this clear
>
> **4) Maybe the supplement should be referenced more often in the main paper since it provides necessary information for understanding the technical details.**
>
> Thanks for pointing this out. We will ensure this.
>
> **5) For reproducibility, the author should consider publishing their code. Also, a comparison of sum-pooling and average-pooling will better justify the choice of the latter.**
>
> We will open-source the training code soon for the benefit of the community.  We had experimented with sum pooling vs average pooling of embeddings on the SEGSalt setting discussed in the paper. We will add these results to the main paper as well
>
> method, test L1 (lower is better), test SSIM(higher is better) , test PSNR (higher is better)
>
> sum, 164.03, 0.46, 16.60
>
> avg, 163.77, 0.47, 16.65
>
>
>
>
> **6) The scalability of the network should be discussed. For example, how many MLPs are needed to cover the gulf dataset in the experiment?**
>
> The MMI-Unscrambler approach to multi-modal data ingestion makes the model scalable to large, complex data such as seismic. We create a billion parameter network of velocity model prediction.
> The details of the model architecture are as follows: (a) we use a one-layer MLP to map each trace signal into 10240-dimensional embedding.  Then, we perform a three-layer MLP on the trace embeddings. Each MLP layer has 4096 hidden units.  (b) For each trace’s 4D location information, we use an MLP to map it into a 512 dimension vector. Then, we concatenate the trace embedding with the location embedding and perform a five-layer MLP with 4096 hidden units in each layer. Finally, we combine the trace information by taking their average and complete the final prediction via one layered MLP. For each MLP layer used in the paper, we perform ReLU activation afterward.  We will add more details on network architecture to the paper.

---

> > ### Comment · Reviewer_2jDV · 2021-08-11
> > **Thanks for your response, but the figures are not changed in the manuscript**
> >
> > Thanks for your response. One quick question is that the figures in the manuscript are not changed. Are the authors planning to incorporate the changes later? or a separate .pdf is created to reflect the edits?

---

> > > ### Author Response · Authors · 2021-08-11
> > > **Revising Figures**
> > >
> > > Thanks for the comment. According to the current policy, we are not allowed to submit paper revisions during the review process. Meanwhile, we checked the console and did not found the portal to upload the revised figures. We will revise the figures following your great suggestions later.

---

### Official Review · Reviewer_bCTP · 2021-07-17

**Rating:** 7
**Confidence:** 2

**Summary:**

The paper proposes Aux-SI approach that handles the large-scale seismic inversion problem via dividing it into small blocks of property models, which makes the training feasible. The authors introduce MMI-Unscrambler architecture to handle the multi-modal information for the task. The empirical study shows that the proposed approach outperforms the state-of-the-art algorithms and is scalable to handle large-scale real-world datasets.

**Limitations And Societal Impact:**

The authors addressed them.

**Main Review:**

The main novelty of the work is to divide the large-scale seismic inversion problem into local inversion problems, each trained based on its context. The potential challenge of this division is that each block is influenced by all traces, and traces in a single block's context will also account for other blocks. The authors explain how their approach handles these challenges well and provides reasonable empirical studies.

Questions/Comments:
1. What does the M(q.z) in line 177 mean? Does q.z mean the depth information of q?
2. Line 118, in problem definition 1 --> defined in Problem 1
3. Line 195, missing space
4. Line 210, for following reason --> for the following reason
5. Line 217. sum-pooling in standard --> sum-pooling is standard
6. line 220, the data prohibit us --> the data prohibits us

**Time Spent Reviewing:**

4

---

> ### Author Response · Authors · 2021-08-10
> **Thank you for the comprehensive review, please see the following clarifications.**
>
> We thank the reviewer for supporting our paper!  We will fix the typos pointed out in points 2-6. Also, to answer point 1, yes, "q.z" refers to the depth of point q. We will clarify this in the paper.

---

### Review · Ethics_Reviewer_K6k1 · 2021-08-11

**Recommendation:**

Yes, concerns are possible to address - please add a sentence to the conclusion, as suggested by reviewer ahiS.

**Ethics Review:**

Reviewer ahiS raises a concern about the application domain for the methods: "Big Oil".  I agree that it can't hurt to add a sentence to the conclusion, reminding the reader about the application domain, as reviewer ahiS suggests.  But I don't see the choice of a domain as a reason to reject the paper.  After all, this is a legitimate domain, not something that's related to criminal activity, say, or to race science.

---

> ### Author Response · Authors · 2021-08-31
> **Addressing the concerns**
>
> Thank you for the review and the support. We will add relevant discussion to the paper and its conclusion as per the suggestion of reviewer ahiS.

---

### Review · Ethics_Reviewer_27SZ · 2021-08-12

**Recommendation:** All I might ask is if there are less …

---

> ### Author Response · Authors · 2021-08-31
> **Less heavy methods**
>
> Thank you for the review and the support. Reducing the computational load of DL especially when applied to large scale machine learning problems such as the one discussed in this paper is an active area of research.  Leveraging the research in this area, we hope to achieve energy efficient solutions to the problem of seismic inversion.

---

### Decision · Program_Chairs · 2021-09-27

**Decision:**

Accept (Poster)

**Comment:**

This work presents a SOTA Multi-layer perceptron solution for seismic inversion. The reviewers all recognized the improved performance relative to past approaches and the design and work involved in creating a practical block-based multi-modal approach. In particular the care taken to acknowledge and address the issues stemming from cross-block correlations. The authors also seemed to have proposed very doable revisions, which would even further strengthen the paper. Therefore I am happy to recommend that this work be accepted at NeurIPS.